# Terson Syndrome: Not to Be Missed in Patients with Disorders of Consciousness

**DOI:** 10.3390/brainsci13060879

**Published:** 2023-05-29

**Authors:** Errikos Maslias, Sergiu Vijiala, Jean-Benoit Epiney, Lazaros Konstantinidis, Aki Kawasaki, Karin Diserens

**Affiliations:** 1Unit of Acute Neurorehabilitation, Department of Clinical Neurosciences, Service of Neurology, Lausanne University Hospital (CHUV), University of Lausanne, 1011 Lausanne, Switzerlandcabinet.epiney@svmed.ch (J.-B.E.); karin.diserens@chuv.ch (K.D.); 2Stroke Center and Department of Clinical Neurosciences, Service of Neurology, Lausanne University Hospital (CHUV), University of Lausanne, 1011 Lausanne, Switzerland; 3Department of Ophthalmology, University of Lausanne, Jules-Gonin Eye Hospital, 1004 Lausanne, Switzerlandaki.kawasaki@fa2.ch (A.K.)

**Keywords:** disorders of consciousness, Terson syndrome, clinical cognitive–motor dissociation

## Abstract

The diagnosis of clinical cognitive motor dissociation (^c^CMD) can be hindered by pitfalls during standardized clinical evaluation based on gold-standard neurobehavioral rating scales. We introduce here a new pitfall, by reporting two cases of Terson syndrome (TS) after subarachnoid haemorrhage (SAH) caused by the rupture of an anterior communicant artery aneurysm, hospitalized in the Acute Neurorehabilitation Unit (ANR) of CHUV. TS is reported to occur in 8–19.3% of patients suffering from SAH. It can lead to significant visual impairment and if unrecognized, may impair the patient’s capacity to interact appropriately with the environment; it thus presents an important pitfall in recognizing clinical cognitive–motor dissociation (^c^CMD) in patients with altered states of consciousness. An early ophthalmological exam should be considered in all patients with SAH and disorders of consciousness or visual complaints.

## 1. Introduction

The thorough clinical assessment of disorders of consciousness (DOC) through neurobehavioral rating scales, such as the Coma Recovery Scale—Revised (CRS-R), remains the gold standard for accurate prognostication. However, many clinical situations can interfere with CRS-R-based clinical evaluation (which requires consistent sensory input and motor output), and can lead to an erroneous diagnosis with serious consequences, especially in patients with clinical cognitive motor dissociation (^c^CMD). A systematic search for such pitfalls, which hinder responses, should be performed during the evaluation of patients with a disorder of consciousness.

We aim to describe the case of two patients presenting with Terson syndrome (TS), which can be an important pitfall in recognizing altered states of consciousness, such as ^c^CMD in patients with SAH, and underlines the importance of systematic screening. TS is not uncommon after a subarachnoid haemorrhage, and is reported to occur in 8–19.3% of cases in observational studies and systematic reviews [1,2]. TS, described by the German ophthalmologist Moritz Litten in 1881 and then by the French ophthalmologist Albert Terson in 1900, is defined as intraocular haemorrhage associated with subarachnoid or intracerebral haemorrhage. The pathophysiology of this syndrome has not been totally elucidated. Many theories have been investigated. One suggests that bleeding arises from the rupture of retinal vessels that follows a sudden surge in intracranial pressure [3], but experimental studies have shown that the intravenous pressures are not high enough to create an intraocular hemorrhage [4]. Another theory suggests that blood originates intracranially, with TS representing an extension of subarachnoid blood into the vitreous humor [5]. Recently, the hypothesis of glymphatic reflux has been proposed [6]. The most common etiology of TS involves subarachnoid hemorrhage due to ruptured cerebral aneurysms, but other causes such as hypertension, tumor, strangulation, and trauma have been reported in the literature [7].

Some authors [8] have suggested that TS is more frequent in patients presenting with subarachnoid hemorrhage, secondary to the rupture of an aneurysm in the anterior circulation; however, further studies [9] have concluded that there are insufficient data for comparative analyses.

We report here two cases of TS after subarachnoid hemorrhage (SAH), both of which had a Fisher grade of 4 caused by the rupture of an anterior communicant artery aneurysm and were hospitalized in the Acute Neurorehabilitation (ANR) Unit at Lausanne University Hospital (CHUV).

## 2. Case Descriptions

Patient 1 is a 63-year-old woman who presented with a severe headache, vomiting, generalized tonic–clonic seizures and a rating of 3/15 on the Glasgow Coma Scale (GCS). The initial cerebral CT scan showed a SAH, with a Fisher grade of 4, caused by the rupture of a 6 × 5 mm anterior communicating cerebral artery aneurysm. The patient was admitted to the Intensive Care Unit (ICU) and required intubation. On initial evaluation, all brainstem reflexes were absent. However, 24 h later, the clinical assessment had evolved favorably, with the reappearance of brainstem reflexes, and the patient underwent an endovascular embolization of the ruptured aneurysm and a placement of an external ventricular drain. Due to the development of intracranial hypertension, a craniotomy was performed. After extubation, 26 days later, a tracheotomy was performed, sedation was withdrawn, and the patient presented a pathological awakening with no response to external stimuli. On first evaluation in the ICU by our mobile ANR team, the patient’s rating on the French version of the Coma Recovery Scale—Revised (CRS-R) was evaluated as 5 (presence of auditory startle, visual fixation, oral reflexive movements and eye opening with stimulation), corresponding to a minimally conscious state. The MBT-r evaluation identified additional positive motor signs, such as a response to noxious stimulation (nail bed and positive nipple sign), confirming the expression of residual cognition. Therefore, clinical cognitive–motor dissociation (^c^CMD) was diagnosed. After further investigations, several pitfalls explaining the ^c^CMD were identified: bi-frontal lesions causing an akinetic mutism, a central pontine and a left middle cerebral artery (MCA) ischemic lesion provoking left hemi-negligence, and a peripheral polyneuropathy impeding movement.

After several weeks, the patient complained of visual deficit. A fundoscopic examination at the bedside performed 44 days post-extubation revealed an extensive vitreous hemorrhage in both eyes. The posterior pole could not be visualized. The visual status could not be determined due to the patient’s poor cooperation. A vitrectomy was performed on the left eye. Postoperatively, the patient was able to visually track a light source, but her poor cognitive status prevented a more precise evaluation of visual function. The bilateral TS was not initially detected upon the first examination of the cerebral CT scan performed 20 days after admission (Figure 1), but was retrospectively confirmed after reviewing the CT images with a senior neuroradiologist. The patient slowly progressed and was discharged to a neurological rehabilitation center after 39 days of hospital stay in the ANR unit.

Patient 2 is a 47-year-old woman who presented with a rapid-onset acute severe headache, followed by a loss of consciousness, generalized tonic–clonic seizures and a rating of 6/15 on the GCS. On admission, she was immediately intubated and transferred to the ICU. The initial cerebral CT scan showed a SAH of Fisher grade 4, caused by the rupture of a 4.5 × 3.3 mm anterior communicating cerebral artery aneurysm with third and lateral ventricular effraction and a left frontal intra-parenchymal hematoma. She underwent the endovascular embolization of the ruptured aneurysm and the placement of an external ventricular drain, which was then replaced by a ventriculoperitoneal drain due to a persistent hydrocephaly. Her ICU stay was complicated by cerebral vasospasms that required treatment with IV nimodipine. The patient was extubated after 28 days and a tracheotomy was performed. After sedation withdrawal, she presented a pathological awakening. She was evaluated in the ICU by the mobile ANR team, and according to the CRS-R, she was classified as having unresponsive wakefulness syndrome (UWS). However, MBT-r assessment testified the presence of a ^c^CMD (defensive response to nail bed noxious stimulation and positive nipple sign). Further investigations identified bi-frontal lesions causing an akinetic mutism, initially explaining the ^c^CMD. Electromyoneurography excluded a peripheral polyneuropathy. Several weeks after, the patient complained of visual deficits. Bedside ophthalmological examination revealed a visual acuity of counting fingers in both eyes, and indirect fundoscopy showed a vitreous hemorrhage in both eyes, confirming TS. Due to a favorable clinical evolution, vitrectomy was not proposed in the acute setting and the patient was discharged to a neurological rehabilitation center. One year after the SAH, her visual acuity was 80% in the right eye and 50% in the left eye.

Baseline patient characteristics are depicted in Table 1.

## 3. Methods and Ethical Considerations

The neuroimaging of choice on admission for both patients was cerebral CT scan (256–detector row Revolution CT, GE Healthcare). The 3 Tesla MRI (Vida, Siemens Healthineers) was used subsequently. A senior neuroradiologist and a senior neurologist or neurosurgeon evaluated all CT- and MRI-based neuroimaging. Bedside ophthalmological examination was performed by an ophthalmologist resident and confirmed by a senior ophthalmologist.

Assessment of CRS-R and MBT-r for both patients was performed or supervised by certified examiners in a non-blinded way. Data were anonymized using the principles of the Health Insurance Portability and Accountability Act (HIPAA) Safe Harbor Privacy Rule [10], conducted in compliance with the Swiss Federal Act on Research involving Human Beings, which waives ethical approval for case reports of less than five patients. The consent of the patient and/or his/her relatives for the re-use of personal data is therefore not required under the current Swiss Research legislation. The anonymized data supporting the findings of this study are available from the corresponding author upon reasonable request.

## 4. Discussion

TS may be unilateral or bilateral and can lead to severe visual impairment. This can block the afferent visual pathway and thus affect the capacity of such patients to interact with their environment, masking residual cognition in patients with unresponsive wakefulness syndrome. Specifically, it can abolish the expression of the ocular “positive” motor signs of residual cognition that are systematically evaluated during the assessment of the CRS-R and MBT-r. Notably, both patients had an abolition of ocular movements (visual fixation and pursuit) during the first bedside evaluations in the ICU setting. Consequently, TS could represent an important pitfall in recognizing clinical cognitive–motor dissociation (^c^CMD) in patients with altered states of consciousness after coma due to SAH. In the ICU setting, where neurologists are challenged to give an early and accurate prognosis of coma recovery in order to guide life-sustaining decisions with a major ethical weight, the recognition of such pitfalls is crucial, as they can hinder the presence of conscious perception, which is strongly correlated with a better prognosis. Additionally, the recognition of such pitfalls and the unmasking of perception permits health professionals to give accurate information regarding prognostication to the next of kin.

Additionally, patients may have cognitive impairment that prevents them from verbalizing visual complaints or complying with visual testing, and therefore the identification of TS might be delayed even more, impeding acute neuro-rehabilitation through visual neurosensory stimulation.

The knowledge of such visual impairment could guide the interdisciplinary ANR team into using adequate strategies in order to stimulate the neuroplasticity of the damaged brain regions, by using alternative sensory pathways and visual compensation techniques.

Furthermore, the delayed management of such a complication can lead to several further complications, such as a long-term serious visual impairment especially if the patient has a vitreous hemorrhage, thus hindering and delaying the rehabilitation process. Therefore, it should be quickly clarified whether the visual impairment results from the brain lesions or from a vitreous hemorrhage associated with a raised intracranial pressure, in order to treat it appropriately for better prognosis. In every patient with suspected TS, a complete dilated funduscopic examination is critical to evaluate the severity of the intraocular hemorrhage. Depending on the opinion of the ophthalmologist, B-scan ultrasonography could be useful for ruling out other pathologies, such as a retinal detachment that may not be visible in the setting of a vitreous hemorrhage; however, this examination could be difficult in the ICU setting. In cases associated with trauma, a posterior vitreous detachment or retinal break also must be ruled out in order to avoid unnecessary interventions. The flowchart below (Figure 2) represents a systematic and structured approach for clinicians to follow in order to appropriately and precisely diagnose altered states of consciousness and guide paraclinical investigations. Between the different confounding factors leading to the misdiagnosis of the patient’s state of consciousness, shown in Figure 2, TS must be suspected and investigated in patients with severe SAH, who do not interact according to the standard neurological scales.

Multiple studies have demonstrated the important role of TS as a predictor of unfavorable outcomes and mortality in patients with SAH [9,12]. Among patients with SAH, those with TS have a 4.8-fold increased risk of death compared to those without a vitreous hemorrhage [9].

The first therapeutical approach before surgery is to assure an elevated head positioning with bed rest and the avoidance of any medications that can promote a vitreous or intraocular hemorrhage, such as anticoagulation or nonsteroidal anti-inflammatory drugs (NSAIDs). The resolution of symptoms may take months and some studies have demonstrated an average of 9 months for the clearance of such hemorrhages. In patients with a significant and persisting visual impairment due to TS, microsurgical vitrectomy with or without laser membranotomy is recommended. The importance of visual efference in the acute phase of the neurorehabilitation of patients with severe brain injury should also be considered in the decision process of early intervention. Patients with TS are usually poor surgical candidates because of the severity of the intracranial hemorrhage. Furthermore, they often do not survive the devastating event and early complications, so the likelihood of an unfavorable outcome should always be considered when deciding upon an ophthalmological intervention.

## 5. Conclusions

In conclusion, TS is a common situation in patients with SAH, especially when severe, and it prevents patients’ interaction with their environment; it thus presents an important pitfall in recognizing ^c^CMD in patients with disorders of consciousness. Since prospective studies have shown a higher frequency of TS than retrospective studies, suggesting that vitreous hemorrhages are not well documented [6], clinicians should be aware of this condition and have a low threshold for performing ophthalmologic assessment. In our opinion, an early ophthalmological exam should be considered in all patients with SAH and disorders of consciousness or visual complaints in order to promptly diagnose TS, and start appropriate neurorehabilitation. More prospective studies are needed in order to determine the incidence of TS, ascertain the associated risks in order to screen patients in the early phase after SAH, determine the prognostic factors for favorable outcomes and reveal its clinical implications in the accurate diagnosis of disorders of consciousness such as ^c^CMD and in the ANR setting of post-SAH patients.

## Figures and Tables

**Figure 1 brainsci-13-00879-f001:**
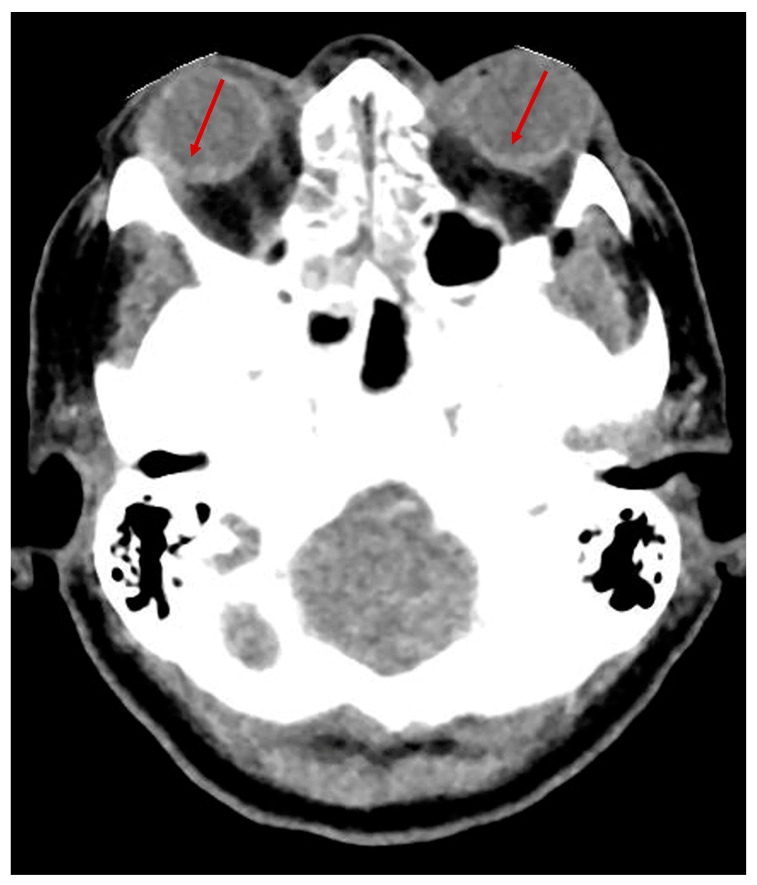
No-contrast brain CT scan of patient 1 showing bilateral Terson’s syndrome (arrows); diagnosis was retrospectively confirmed by a senior neuroradiologist.

**Figure 2 brainsci-13-00879-f002:**
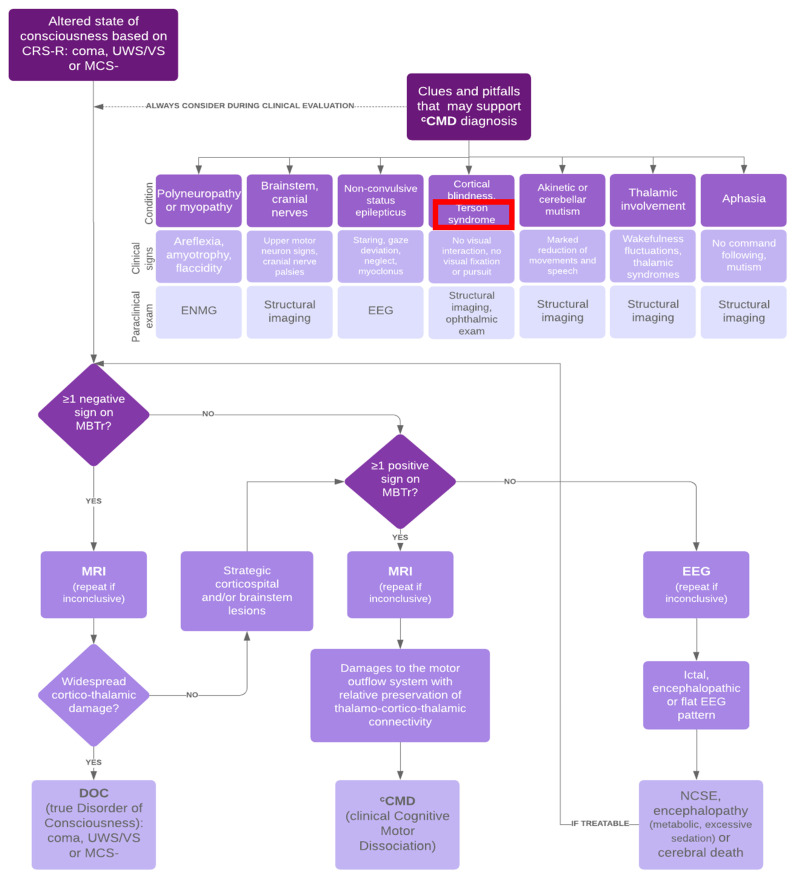
Acute assessment of patients with major cerebral impairment. Adapted from [11]. CRS-R—Coma Recovery Scale—Revised; UWS—Unresponsive Wakefulness Syndrome; VS—Vegetative State; MCS—Minimal Conscious State; ^c^CMD—clinical Cognitive Motor Dissociation. ENMG—electroneuromyography. EEG—Electroencephalogram; MBT-r—Motor Behavior Tool—Revised; MRI—Magnetic Resonance Imaging. DOC—Disorder of Consciousness. NCSE—Nonconvulsive Status Epilepticus.

**Table 1 brainsci-13-00879-t001:** Baseline patient characteristics.

Variable	Patient 1	Patient 2
Age (years)	63	47
Sex	Female	Female
Ethnicity	White	White
Clinical exam		
CRSR scale within 24 h of admission	7/23 (MCS−)	17/23 (MCS+)
CRSR scale at 7 days	8/23 (MCS−)	23/23 (EMCS)
CRSR scale at hospital discharge	13/23 (MCS+)	23/23 (EMCS)
MBT at 24 h	^c^CMD	^c^CMD
Perception signs on first examination (24 h after stop of sedation) according to the MBT-r	Response to a noxious stimulation (positive nipple sign, nail bed)	Visual fixation or pursuit Response to a noxious stimulation (positive nipple sign, nail bed and grimace)
Neuroimaging on admission	Cerebral CT scan	Cerebral CT scan
Ophthalmological examination findings	Bilateral Terson’s syndrome	Bilateral Terson’s syndrome
Treatment of TS	Bilateral vitrectomy	Conservative management
Duration of ICU stay (in days)	31	38
Duration of stay in the Acute Neurorehabilitation Unit (in days)	37	24
Discharged orientation	Neurorehabilitation center	Neurorehabilitation center

CRS-R—Coma Recovery Scale—Revised. MCS—Minimal Conscious State; EMCS—emergence from MCS. MBT-r—Motor Behavior Tool—Revised. ^c^CMD—clinical Cognitive Motor Dissociation. ICU—Intensive Care Unit.

## Data Availability

The anonymized data of this study are available from the authors upon reasonable request.

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
