# Peer review of "Terson Syndrome: Not to Be Missed in Patients with Disorders of Consciousness"

_brainsci, 2023, doi:10.3390/brainsci13060879_

Round 1

Reviewer 1 Report

In this case report, the authors presented two cases of Terson Syndrome after subarachnoid haemorrhage. This case study is interesting and well-presented. The language of the manuscript should be further revised and all abbreviations should be defined (e.g. ICP line 45 and ENMG line 101). Additionally, all the abbreviations used in figure 2 should be defined in the figure legend.

Moderate revision of the language is required.

Author Response

We strongly appreciate the reviewer’s feedback and comment.

We have now defined all abbreviations as follows:

  • ICP: intracranial pressure
  • ENMG: electromyoneurography
  • Abbreviations of figure 2: CRS-R=Coma Recovery Scale-Revised. UWS=unresponsive wakefulness syndrome. VS=vegetative state. MCS=Minimal Conscious State. cCMD=clinical Cognitive Motor Dissociation. ENMG=electroneuromyography. EEG=electroencephalogram. MBT-r=Motor Behavior Tool–revised. MRI=Magnetic Resonance Imaging. DOC=Disorder of Consciousness. NCSE=Nonconvulsive status epilepticus.

The language of the manuscript has now been further revised by a native English speaker, and changes have been implemented in red.

Reviewer 2 Report

This case report teaches the lesson that TS may be missed in SAH cases with impaired consciousness, but I do not see any further value in this case report.   Figure 1.: Why should you dare to present an image that is not recognized as TS? Minor editing of English language should be required.

Author Response

Thank you for your comment.

We aimed to describe the importance of Terson syndrome, as a pitfall that can hinder clinical cognitive motor dissociation (cCMD).

Figure 1 illustrates a non-contrast brain CT scan showing bilateral Terson’s syndrome; crescentic hyperdensity, relative to the vitreous humor, in the posterior globe, which in association with subarachnoid heamorrhage are highly suggestive of this diagnosis. We have now put the arrows in place in order to highlight this radiological finding, which was then confirmed by funduscopic examination which revealed an extensive vitreous haemorrhage in both eyes.

The language of the manuscript has now been further revised by a native English speaker, and changes have been implemented in red

Reviewer 3 Report

Thank you for inviting me to review this interesting paper, presenting 2 cases of patients who suffered from the comorbidity of subarachnoid haemorrhage (SAH) followed by Terson syndrome (TS). According to the paper, TS is reported to occur in a cosiderable percentage (8-19.3%) of patients suffering from SAH. It can lead to significant visual impairment and if unrecognized, may impair the patient’s capacity to interact appropriately with the environment and presents thus an important pitfall recognizing clinical cognitive-motor dissociation (cCMD) in patients with altered states of consciousness. 

I think the paper is well written and should be published.

The only comment I have is that the figures need to be lightly modified:  in figure 1, the arrows are not in place and in figure 2, the red box in not exactly in place.

Good luck!

Author Response

We strongly appreciate the reviewer’s positive feedback.

We have now modified the figures in order for the arrows to be in place in figure 1 and the box in place in figure 2.

Round 2

Reviewer 2 Report

I understand the authors' argument.

Please state in the text and figure legends that you have retrospectively reviewed the CT images and found the findings of TS.

Author Response

We thank the reviewer for his positive feedback and comments. 

We have now adapted the text as follows:

  • Figure 1. No contrast brain CT scan of patient 1 showing bilateral Terson’s syndrome (arrows); diagnosis was retrospectively confirmed by a senior neuroradiologist.
  • Text - description patient 1: The bilateral TS was not initially detected on the first examination of cerebral CT scan performed 20 days after admission (Figure 1) but was retrospectively confirmed after reviewing the CT images with a senior neuroradiologist.

We hope that match the reviewers’ expectations.